# Pain and Disability Reduction Following Rib Manipulation in a Patient Recovering from Osteomyelitis of the Thoracic Spine

**DOI:** 10.3390/healthcare13121355

**Published:** 2025-06-06

**Authors:** Joshua Prall, James Dunning, Ian Young, Michael Ross, James Escaloni, Paul Bliton

**Affiliations:** 1American Academy of Manipulative Therapy Fellowship in Orthopaedic Manual Physical Therapy, 445 Dexter Avenue, Suite 4050, Montgomery, AL 36104, USA; drjamesdunning@gmail.com (J.D.); tybeewellness@gmail.com (I.Y.);james.escaloni@spinalmanipulation.org (J.E.); paul.bliton@spinalmanipulation.org (P.B.); 2Montgomery Osteopractic Physical Therapy & Acupuncture, Montgomery, AL 36104, USA; 3Tybee Wellness & Osteopractic, Tybee Island, GA 31328, USA; 4Department of Physical Therapy, Daemen University, 4380 Main Street, Amherst, NY 14226, USA; mross@daemen.edu; 5Wellward Regenerative Medicine, Lexington, KY 40509, USA; 6United States Department of Veterans Affairs, William Middleton VA Hospital, Madison, WI 53705, USA

**Keywords:** spinal manipulation, osteomyelitis, mobilization, physical therapy

## Abstract

**Introduction:** Spinal thrust manipulation has been found useful for improving pain and mobility in musculoskeletal conditions of the thoracic spine. This case report highlights the importance of incorporating high-velocity low-amplitude (HVLA) thrust manipulation to the mid-thoracic rib articulations in a patient experiencing thoracic spine pain associated with an acute onset of osteomyelitis at levels T7–T9. **Detailed Case Description:** A 49-year-old female who was recovering from osteomyelitis of the thoracic spine 4 months prior was referred to physical therapy by her neurosurgeon. Her osteomyelitis infection resulted in a bone-on-bone interaction between T7 and T9, resulting in significant thoracic spine pain. Severe restrictions in active range of motion (AROM) were found in extension and right and left rotation. At initial evaluation, the patient’s pain intensity score was 8/10 (NPRS, 0–10), the disability score was 46/50 (NDI, 0–50), and the patient-specific functional scale score was 3/10 (PSFS, 0–10). Initially, interventions included grades I-IV posterior to anterior (PA) mobilizations of the thoracic spine from levels T2 to T9, mobilization with movement of the thoracic spine for extension and rotation bilaterally, scapular stabilization, and thoracic mobility exercises. Treatment progressed to HVLA thrust manipulation techniques targeting the costotransverse articulations of ribs 2–9. **Discussion:** Following the initial eight treatment sessions over 4 weeks, minimal improvement was observed for pain (NPRS from 8/10 to 6/10), disability (NDI from 46/50 to 34/50), and thoracic extension AROM (13°). However, during visits 9–16, the addition of HVLA thrust manipulation targeting the costotransverse articulations resulted in significant improvements in pain, disability, and AROM. The patient was subsequently discharged after 16 visits and able to return to a full workday as a teacher without any thoracic pain or ROM restrictions. At the 6-month follow-up, the patient outcomes remained, and she was working with no restrictions. **Conclusion:** The addition of HVLA thrust manipulation targeting the mid-thoracic rib articulations to a program of non-thrust mobilization and exercise appeared useful for improving pain, disability, and range of motion in a patient recovering from osteomyelitis of the thoracic spine.

## 1. Introduction

Osteomyelitis is an inflammatory condition that affects the bones and is frequently the result of an infection within the body. Osteomyelitis can be acute, sub-acute, or chronic, with *Staphylococcus aureus* being the catalyst 70–80% of the time [1,2,3]. Osteomyelitis that develops in the spine, or vertebral osteomyelitis, is a rare condition that affects 4.2 out of every 100,000 people [1,2,3,4]. Vertebral osteomyelitis of the thoracic spine can often lead to bony deformities due to infection infiltrating the spinal canal [5,6]. Osteomyelitis in the thoracic spine can be difficult to initially diagnose and can easily be clinically mismanaged due to the vague symptoms in the thoracic region during the initial examination [4,5,7,8]. In cases of vertebral osteomyelitis, the sequalae of events includes thoracic spine pain, hypomobility, weakness, and limited range of motion [5,9,10]. Typically, patients diagnosed with this condition do not require surgery and will clear the infection with intravenous antibiotic therapy [11,12,13,14,15,16].

Common interventions for thoracic spine pain and dysfunction are therapeutic exercise, motor control training, patient education, and manual therapy techniques [17,18,19,20,21,22,23]. Quality evidence is lacking to support the effect of exercise and manual therapy as standalone treatments for improving pain and disability in the thoracic spine region [22]. However, evidence generating the largest improvements in pain and function in patients with thoracic spine dysfunction comes from multimodal interventions, including a combined approach of mobility exercises, manual therapy, motor control training, and patient education [21,22,23]. Manual therapy techniques directed to the thoracic spine that are commonly used include graded non-thrust mobilizations, non-thrust mobilizations with movement, and high-velocity low-amplitude (HVLA) thrust manipulation [21,22,23,24].

HVLA thrust manipulation targeting the thoracic spine can improve pain, range of motion, and disability for patients with thoracic spine-related pain [21,22,23]. To date, there is a paucity of research on the effects of costotransverse joint manipulation for thoracic spine pain secondary to an acute onset of osteomyelitis when traditional manual therapy techniques are yielding minimal to no effect on patient-reported outcome measures of pain and disability. This case report discusses the interventions and outcomes in a patient with upper to mid-thoracic spine pain secondary to a bony deformity caused by the clinical sequalae of osteomyelitis in the thoracic spine. Specifically, this case report highlights improvement in pain, range of motion, strength, and disability outcome measures following the use of bilateral joint manipulation targeting the costotransverse joints at levels T2–T9.

## 2. Detailed Case Presentation

A 49-year-old female was referred to physical therapy by her neurosurgeon with a diagnosis of upper back pain secondary to osteomyelitis. This resulted in a bone-on-bone interaction between T7 and T9 and subsequent pain and movement restrictions. Her past medical history included hypertension, hyperlipidemia, anxiety, and recent onset of insidious osteomyelitis. The patient’s past medical history, surgical history, and medications are listed in Table 1. Magnetic resonance imaging (MRI) and radiograph showed the presence of osteomyelitis in December of 2023. The patient was treated with intravenous (IV) antibiotics for 5 weeks, and the patient received in-patient physical therapy but refused all narcotic pain medication. At the conclusion of the 5-week IV treatment, bloodwork and repeat imaging showed that the osteomyelitis infection had cleared. The patient was able to start physical therapy immediately following the clearance of the infection. She was seen for 16 visits over the course of 8 weeks and a 6-month follow-up post discharge. No adverse events occurred during the course of her treatment.

Prior to the examination, informed consent was obtained, and the patient’s rights were protected. On examination, the patient reported consistent upper back pain for 4 months that had occurred after developing, being diagnosed, and treated for osteomyelitis of the thoracic spine at levels T7–T9. There were no complaints of extremity pain or radicular symptoms in the upper or lower extremities (Figure 1, Table 2). The pain pattern she was experiencing included many potential pain sources, drawing attention to specific differential diagnoses (Figure 2). The patient’s pain intensity score was 8/10 (NPRS, 0–10), the disability score was 46/50 (NDI), 0–50), and the functional scale score was 3/30 PSFS, (0–10) [25,26,27,28]. The patient was hoping to return to work as a high school teacher without pain.

### 2.1. Differential Diagnosis

A neurological examination demonstrated normal (i.e., 2+) deep tendon reflexes for the biceps (C5), brachioradialis (C6), and triceps (C7). Hoffmann’s, Babinski, and clonus were negative bilaterally, suggesting no sign of upper motor neuron lesion. Upper quarter and lower quarter screenings were performed due to the patient’s complaint of pain being located between T2 and T9. Dermatome testing for both upper and lower quarter screening demonstrated normal sensation. Myotome testing for both upper and lower quarter screening also demonstrated normal strength for all associated nerve roots. The seated slump test and the passive neck flexion test were both performed but were unable to reproduce the patient’s symptoms. Gait analysis revealed a non-ataxic gait pattern and was visually performed by the physical therapist in a straight hallway in the clinic. Neurological testing suggested that the thoracic spine pain was not related to a nerve root or spinal cord lesion.

Cervical spine ROM bilaterally was within functional limits (WFL) for flexion, extension, bilateral side-bending, and bilateral rotation. The cervical spine strength test revealed 4/5 manual muscle testing (MMT) for all cervical spine movements (Table 3). Neural tension testing of the cervical spine included a seated slump test and passive neck flexion test, which were both negative (Table 4). Posterior to anterior (PA) mobilizations centrally and unilaterally to the cervical spine in prone position were painless with no reproduction of symptoms.

Thoracic spine ROM was severely limited in extension with a subsequent overcompensation of thoracic flexion. Bilateral side-bending and bilateral rotation were also both limited. Active thoracic extension, bilateral side-bending, and bilateral rotation reproduced the patient’s symptoms at levels T7–T9 (Table 5). The patient’s strength test revealed 2/5 manual muscle testing (MMT) for all thoracic spine movements (Table 3). The thoracic compression test was positive and reproduced the patient’s symptoms at levels T7–T9. PA mobilizations both centrally and unilaterally revealed significant hypomobility and reproduced the patient’s pain at levels T2–T9. The examination results were consistent with significant hypomobility and pain due to a recent onset of osteomyelitis at levels T7–T9.

### 2.2. Treatment

Initially, the patient’s treatment focused on therapeutic exercise to improve thoracic mobility and motor control of the scapular stabilizers, and later, HVLA thrust manipulation was introduced to target the costotransverse articulations. The HVLA thrust manipulation technique included the patient lying in a supine position with an anterior to posterior force directed towards the physical therapist’s hand that was in contact with the costotransverse joint of levels T7–T9 on the posterior aspect of the thorax. The patient was seen for 16 visits over a course of 8 weeks, which consisted of two sessions per week for 8 weeks (Table 6).

During visits 1 through 4, the focus was on mobility and pain control. Motor control exercises consisted of cervical retractions for deep neck flexor (DNF) strengthening and scapular retractions, which were both initiated in the clinic. A home exercise program (HEP) consisting of mobility exercises for the thoracic spine, the early initiation of scapular stabilization, and DNF strengthening was implemented. Grades I and II PA mobilizations directed centrally and unilaterally were performed for levels T2–T9 for pain control and joint nourishment (Table 6).

During visits 5 through 8, the patient’s pain started to subside, and she was able to progressively gain the ability to perform thoracic extension, bilateral rotation, and bilateral side-bending with the addition of the mobilization of movement techniques for thoracic extension, thoracic rotation, and thoracic side-bending. Motor control exercises were able to be progressed to include resistance and functional movement patterns for scapular stabilization. The patient was re-evaluated after her eighth visit, and there were only minimal improvements in pain (NPRS), disability (NDI), function (PSFS), and mobility with the current treatment model (Table 6) [25,26,27,28].

During visits 9 through 12, the addition of HVLA thrust manipulation targeting the costotransverse articulations was incorporated into the patient’s plan of care due to a lack of significant improvement in the mobility of the thoracic spine during the first eight sessions. Notably, the patient’s osteomyelitis infection had been cleared by MRI, X-ray, and bloodwork; therefore, manipulation was not contraindicated at this point in time. Following HVLA thrust manipulation targeting the costotransverse articulations associated with levels T2–T9 bilaterally, significant improvements in thoracic extension, bilateral thoracic rotation, and bilateral thoracic side-bending mobility were demonstrated. Furthermore, meaningful reductions in the patient’s pain (NPRS) scores were observed following the utilization of HVLA thrust manipulation (Table 6) [25].

During the last four visits, the focus of the program was on mastering functional activities, improving ergonomics for her teaching job, and resolving pain that she was originally experiencing during activities of daily living (ADLs). On discharge, the patient demonstrated clinically meaningful improvements in pain (NPRS), disability (NDI), function (PSFS), and mobility (ROM) (Table 6 and Table 7) [25,26,27,28].

### 2.3. Outcomes and Follow-Up

The patient was discharged from physical therapy after 16 visits over a course of 8 weeks. During her 16th visit, she reported a reduction in pain from a score of 8/10 to 1/10 on the NPRS, which exceeded the minimally clinically important difference for the shoulder (two points) [25]. Her NDI score was reduced from 46/50 to 22/50, and her PSFS score increased from 3/30 to 21/30, both exceeding the minimally clinically important differences of 7.5 for the NDI and 2–3 points of the PSFS per section, respectively [26,27,28]. Cervical active and passive range of motion was within functional limits for all motions. Active and passive thoracic spine range of motion improved dramatically after the initiation of costotransverse joint manipulation. On discharge, her ROM and strength improved to the point where the patient was able to perform activities of daily living with a 1/10 pain score on NPRS on occasion (Table 7 and Table 8). Notably, the patient was also able to work an entire shift as a teacher without having to take a break or call off work due to pain. The patient had followed up with the clinic 1 month after being discharged to inform the therapist that she continues to work with minimal to no pain and is continuing her home exercise program, which was assessed during her recall of the exercises. During her 6-month follow-up, the patient called the clinic to report her continued high activity level and no pain.

## 3. Discussion

To the authors’ knowledge, this is the first case study to describe the use of HVLA thrust manipulation targeting the costotransverse joints in a patient who was unresponsive to non-thrust mobilization targeting the thoracic spine. The improved self-report pain and disability outcome measures and range of motion suggest that the addition of costotransverse joint manipulation may serve as an important intervention in patients who are unresponsive to non-thrust graded unilateral and central PA mobilizations.

Thoracic spine non-thrust joint mobilization has been suggested to have a multitude of effects on pain, disability levels, muscle force production, and range of motion [21,32,33,34,35,36]. These neurophysiological responses can play a significant role in the rehabilitation of a patient with these limitations [26,34]. The thoracic spine region can be difficult to treat as there are currently no clinical practice guidelines for thoracic spine or upper back pain like there are for low back and neck pain [37,38]. Although the effects of HVLA thrust manipulation targeting specifically the costotransverse joints have not been addressed in prior studies, two prior studies have reported positive effects on pain, disability, and the function of the thoracic spine following costotransverse joint manipulation in patients diagnosed with costochondritis [39,40]. Nevertheless, due to the proximity to the zygapophyseal joints of the thoracic spine, costotransverse joint manipulation can likely play an important role in decreasing pain, improving disability, increasing joint range of motion, improving overall posture, and increasing functional capacity in patients with thoracic spine-related pain and movement dysfunctions [41]. This may be especially useful when thoracic spine mobilization and manipulation are ineffective or have a deleterious effect, as seen in this case due to the patient’s past medical history of recent osteomyelitis at levels T7–T9.

Scapular stabilization to improve strength in the thoracic spine region was also a crucial intervention in this case. Facilitating the strengthening of the rhomboids, middle and lower trapezius, and serratus anterior bilaterally may have further played a role in decreasing pain and improving functional improvement for this patient [42,43,44]. An additional factor that may have aided in her success was a strong therapeutic alliance between the patient and physical therapist. This allowed for increased adherence to her overall rehabilitation program and her home exercise program [45].

HVLA must be performed with caution in patients who have experienced osteomyelitis due to the weakening of the bone tissue, which can increase the risk of a serious adverse event, such as a fracture [46]. The authors recommend conducting a thorough evaluation of the thoracic spine region prior to HVLA thrust manipulation to the impaired area. Once determined to be a safe technique, HVLA can benefit the patient from biomechanical and neurophysiological perspectives. Biomechanically, a reduction in joint adhesions and improved motion have been noted with HVLA thrust techniques targeting the spine. Neurophysiological mechanisms such as pain modulation and a reduction in muscle spasms and hypertonicity have also been noted [47].

Despite the positive outcome HVLA thrust manipulation had when added to a program of scapular stabilization and exercise, several limitations are evident. As a single-subject case report, the findings are inherently limited in generalizability and cannot establish causality between rib manipulation and the observed improvements in this case. Future research conducted using randomized controlled study designs with larger sample sizes is required to better determine the effectiveness of rib manipulation in managing thoracic spine pain and disability.

## 4. Conclusions

Costotransverse joint manipulations may provide improvements when central or unilateral posterior to anterior joint mobilizations or thoracic manipulations do not provide the desired effect. Costotransverse joint manipulations may be incorporated in patients initially experiencing hypomobility and pain in the thoracic spine who are non-responsive to non-thrust mobilizations or mobilizations with movement.

## Figures and Tables

**Figure 1 healthcare-13-01355-f001:**
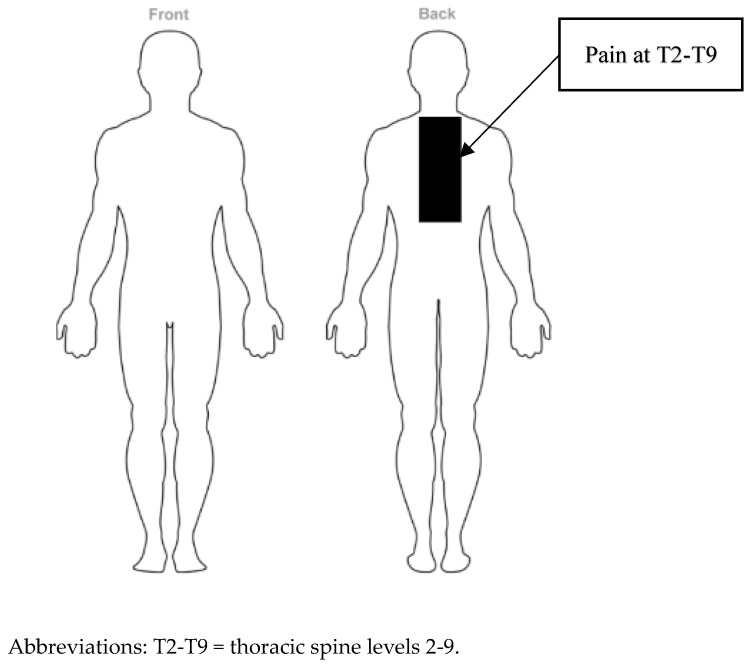
Body chart pain diagram.

**Figure 2 healthcare-13-01355-f002:**
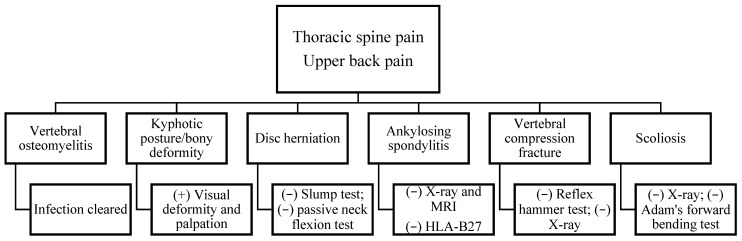
Differential diagnosis.

**Table 1 healthcare-13-01355-t001:** Medical history, surgical history, and medications.

	Description
**Medical history**	Osteomyelitis Osteoarthritis Sacroiliac joint pain Hypertension
**Surgical history**	Multiple root canals
**Medications**	OxyContin (20 mg) prn for pain Norvasc (5 mg) 1 time daily Ceftriaxone (1–2 g) IV once daily Daptomycin (4 mg/kg) IV every 24 h for 7 days

Abbreviations: prn = as needed; IV = intravenous.

**Table 2 healthcare-13-01355-t002:** Symptom location, pain description, and aggravating and easing factors during initial evaluation.

Symptom Location	Pain Description	Aggravating Factor	Easing Factor
Upper and middle thoracic spine (T2–T9)	NPRS severity: 8 Irritability: mild Stage: sub-acute Stability: worsening	Standing > 15 min Sleeping in supine Walking > 30 min Donning and doffing clothing	Sitting with exaggerated thoracic flexion Side-lying position

Abbreviations: T2–T9 = thoracic spine levels 2–9; ≥ greater than.

**Table 3 healthcare-13-01355-t003:** Cervical and thoracic active and passive range of motion (AROM and PROM) and strength assessment during initial evaluation.

Cervical Spine	AROM	PROM	Strength
Flexion	80°	86°	4/5
Extension	72°	75°	4/5
Right side-bending	40°	44°	4/5
Left side-bending	42°	46°	4/5
Right rotation	85°	88°	4/5
Left rotation	80°	84°	4/5
**Thoracic Spine**	**AROM**	**PROM**	**Strength**
Flexion	60°	66°	2/5
Extension	5°	7°	2/5
Right side-bending	18°	22°	2/5
Left side-bending	17°	20°	2/5
Right rotation	12°	15°	2/5
Left rotation	16°	20°	2/5

Abbreviations: AROM = active range of motion; PROM = passive range of motion.

**Table 4 healthcare-13-01355-t004:** Descriptions of special tests.

Test	Description	Positive Test Result
Slump test [29]	The patient flexes the spine and shoulders while the examiner holds the chin and head erect. The patient is asked whether any symptoms are produced. If no symptoms are produced, the examiner flexes the patient’s neck and holds the head down with shoulders slumped to see whether symptoms are produced. If no symptoms are produced, the examiner passively extends one of the patient’s knees to see whether symptoms are produced. If no symptoms are produced, the examiner then passively dorsiflexes the foot of the same leg to see if symptoms are produced.	Reproduction of the patient’s symptoms indicates a positive test, implicating neural tension
Passive neck flexion test [30]	The patient actively performs an upper cervical nod. The examiner passively flexes the lower cervical spine.	A reproduction of pain or other neural symptoms in the thoracic spine is a positive test result
Adam’s forward bending test [31]	The patient needs to bend forward, starting at the waist until the back comes in the horizontal plane, with the feet together, arms hanging, and the knees in extension. The palms are held together.	An asymmetry is observed on one side
Reflex hammer test [29]	The patient is seated, and the examiner taps over each spinous process to see whether pain or muscle spasm is provoked.	If pain or muscle spasm is provoked, a fracture is possible

**Table 5 healthcare-13-01355-t005:** Thoracic active ROM and pain assessment during initial evaluation.

Thoracic Movement	AROM	Pain Location	NPRS
Flexion	60°	Central, T2	2/10
Extension	5°	Central, T5–T9	8/10
Right side-bending	18°	Right unilateral, T5–T9	6/10
Left side-bending	17°	Left unilateral, T5–T9	6/10
Right rotation	12°	Right unilateral, T2–T9	4/10
Left rotation	16°	Left unilateral, T2–T9	4/10

Abbreviations: AROM = active range of motion; T2 = thoracic spine level 2; T5–T9 = thoracic spine levels 5–9; T2–T9 = thoracic spine levels 2–9; NPRS = Numeric Pain Rating Scale.

**Table 6 healthcare-13-01355-t006:** Treatment progression throughout care plan.

Visit	Treatment Plan
1–4	**Mobility**: Thoracic extension, side-bending, and rotation AROM in sitting position, side-lying rotational AROM, doorway stretch **Motor control**: Scapular squeezes in sitting position **Manual therapy**: Grades 1–4 central and unilateral PA mobilizations to reduce pain at levels T2–T9; STM to thoracic paraspinals to reduce muscle tightness; pectoralis minor STM to reduce muscle tightness
5–8	**Mobility**: Continued mobility from visits 1–4 with progressions of ½ foam roll extensions over chair with arms crossed; begin quadruped unilateral rotation exercises; downward dog; tabletop stress with physioball **Motor control**: Continued exercises from visits 1–4; progressed to serratus anterior strengthening, and resistance exercises targeting scapular stabilizers **Manual therapy**: Grades 3–4 central and unilateral PA mobilizations at levels T2–T9 to improve thoracic mobility; mobilization with movement for thoracic rotation, side-bending, and rotation; thoracic manipulation to T5–T7
9–12	**Mobility**: AROM exercises from sessions 1–4 discharged to HEP; progressed to quadruped rotation AROM **Motor control**: Scapular stabilization including resistance bands and dumbbells **Manual therapy**: Continued techniques from visits 5–8; initiation of costotransverse joint manipulation bilaterally from T2 to T7
13–16	**Mobility**: Discharged to patient’s HEP **Motor control**: Scapular stabilization including resistance bands and dumbbells until failure; functional activities including farmers carries, crate carries, and sled pushing and pulling **Manual therapy**: Costotransverse joint manipulation bilaterally from T2 to T7

Abbreviations: PA = posterior to anterior; AROM = active range of motion; HEP = home exercise program; T2–T9 = thoracic spine levels 2–9; T5–T7 = thoracic spine levels 5–7; T2–T7 = thoracic spine levels 2–7.

**Table 7 healthcare-13-01355-t007:** Outcome measures and pain ratings throughout care plan.

Assessment	NPRS	NDI	PSFS
Initial evaluation	8/10	46/50	3/30
Re-evaluation (Week 4)	6/10	34/50	9/30
Discharge (Week 8)	1/10	22/50	21/30

Abbreviations: NPRS = Numeric Pain Rating Scale; NDI = Neck Disability Index; PSFS = Patient-Specific Functional Scale.

**Table 8 healthcare-13-01355-t008:** Cervical and thoracic active and passive range of motion (AROM and PROM) and strength assessment at re-evaluation and discharge.

Cervical Spine (Re-Evaluation)	AROM	PROM	Strength
Flexion	88°	90°	4/5
Extension	74°	81°	4/5
Right side-bending	45°	49°	4/5
Left side-bending	46°	50°	4/5
Right rotation	86°	88°	4-/5
Left rotation	85°	86°	4-/5
**Thoracic Spine** **(Re-Evaluation)**	**AROM**	**PROM**	**Strength**
Flexion	WFL	WFL	2+/5
Extension	11°	15°	2+/5
Right side-bending	22°	25°	2+/5
Left side-bending	25°	27°	2+/5
Right rotation	20°	23°	2+/5
Left rotation	20°	22°	2+/5
**Cervical Spine** **(Discharge)**			
Flexion	WFL	WFL	4+/5
Extension	WFL	WFL	4+/5
Right side-bending	WFL	WFL	4+/5
Left side-bending	WFL	WFL	4+/5
Right rotation	WFL	WFL	4/5
Left rotation	WFL	WFL	4/5
**Thoracic Spine** **(Discharge)**			
Flexion	WFL	WFL	4/5
Extension	35°	40°	3+/5
Right side-bending	40°	46°	3+/5
Left side-bending	35°	39°	3+/5
Right rotation	50°	55°	3+/5
Left rotation	45°	50°	3+/5

Abbreviations: AROM = active range of motion; PROM = passive range of motion.

## Data Availability

There is no data or statistical analysis to share.

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
