# Peer review of "Pain and Disability Reduction Following Rib Manipulation in a Patient Recovering from Osteomyelitis of the Thoracic Spine"

_healthcare, 2025, doi:10.3390/healthcare13121355_

Round 1
Reviewer 1 Report
Comments and Suggestions for Authors
the manuscript by Prall et al. is a case report of thoracic spine pain after osteomyelitis.
It based on the idea that rib manipulation may reduce spine pain. This is an actual topic, but while the idea is interesting, there are few points of concern that would need to be addressed before the manuscript could be suitable for publication.
I think that some corrections could improve the quality of the article.
1) title is appropriate
2) i don't understand if the PT treatment was proposed during antibiotic therapy and if pain medication were administered
3) in line 50-51 the sentence needs to be improved
4) in line 49, when education is cited, it useful could be add this ref How does semantic pain and words condition pain perception? by Lena et al , 2022), for better clarify the therapeutic role of language in pain management
5) The cases is well presented and results are clear.
6) I feel that conclusions are consistent Discussion is exhaustive.
Author Response
Comment#1: Title is appropriate
Response#1: Thank you
Comment#2: I don't understand if the PT treatment was proposed during antibiotic therapy and if pain medication were administered
Response#2: Thank you for pointing this out. I have revised that section of the paper to include the information. In-patient hospital PT was performed at that time but no pain medication was used due to patient refusing narcotic pain medication. It is highlighted in yellow.
Comment#3: In line 50-51 the sentence needs to be improved
Response#3: Thank you. I revised that sentence to improve it's reading quality and it is highlighted in yellow.
Comment#4: In line 49, when education is cited, it useful could be add this ref How does semantic pain and words condition pain perception? by Lena et al , 2022), for better clarify the therapeutic role of language in pain management
Response#4: Thank you. This article has been added to that sentence and updated in the reference list. It is highlighted in the reference list in yellow for your convenience.
Comment#5: The cases is well presented and results are clear
Response#5: Thank you
Comment#6: I feel that conclusions are consistent Discussion is exhaustive.
Response#6: Thank you.
Reviewer 2 Report
Comments and Suggestions for Authors
Dear Authors,
The study's objective is valuable, and the topic is very interesting. You have detailed case descriptions with clear outcome measures and a good and logical treatment progression based on patient response. Also, you have included follow-up data that supports the long-term effectiveness. However, some points need to be addressed, as mentioned below:
Title:
The title is clear and specific
Abstract:
The abstract explains the contents of the manuscript.
Introduction:
Good
Please provide the meaning of abbreviations the first time they are used.
Although you mentioned HVLA in the abstract, you should have mentioned its meaning the first time you mentioned it in the introduction.
Other abbreviations were clearly mentioned in a good way.
Case presentation:
You mentioned your patient's adherence to the home exercise program. How do you measure or quantify her adherence?
Explicitly state whether any adverse events occurred during treatment.
Discussion:
While the authors note the infection was cleared before manipulation, they do not address potential risks of HVLA in post-osteomyelitis patients (e.g., structural fragility, recurrence risk, etc). Please include a paragraph on safety considerations for HVLA in patients with a history of spinal infections, even after clearance, and discuss potential biomechanical or neurophysiological mechanisms of HVLA in post-osteomyelitis cases.
The role of scapular stabilization exercises and therapeutic alliance is emphasized, but their contributions to outcomes are not parsed.
Also, the contributions of HVLA versus scapular stabilization exercises should be differentiated from the observed improvements.
Consider adding a "Lessons Learned" section to guide future case reports or studies.
Give us an overview of the weaknesses of your case report.
Conclusion: Good
References:
Well-referenced with relevant literature.
Number references in the order in which they are first mentioned in the text. Please arrange the references in the correct format.
Tables and Figures:
Tables and figures enhance data interpretation.
Figure 1 (Graphic 1) is a repetition of Table 6; you should keep either Table 6 or the figure. I suggest keeping the table.
Comments on the Quality of English Language
This case report provides valuable insights into the potential utility of HVLA thrust manipulation for thoracic spine dysfunction post-osteomyelitis. The writing is clear, and the clinical reasoning is sound. Addressing the above recommendations would enhance the manuscript’s scholarly impact and clinical relevance.
Author Response
Comment#1: The title is clear and specific
Response#1: Thank you
Comment#2: The abstract explains the contents of the manuscript
Response#2: Thank you
Comment#3: Please provide the meaning of abbreviations the first time they are used. Although you mentioned HVLA in the abstract, you should have mentioned its meaning the first time you mentioned it in the introduction
Response#3: Thank you for pointing this out. The meaning has been added to the manuscript before using the term "HVLA" and it has been highlighted within the manuscript for your convenience.
Comment#4: You mentioned your patient's adherence to the home exercise program. How do you measure or quantify her adherence?
Response#4: Thank you for pointing this out. Adherence was measured during the subject portion of each visit through patient recall. This has been added to the manuscript and highlighted in yellow for your convenience.
Comment#5: Explicitly state whether any adverse events occurred during treatment.
Response#5: Thank you for this addition. It has been added to the introduction of the detailed case presentation and has been highlighted in yellow for your convenience.
Comment#6: While the authors note the infection was cleared before manipulation, they do not address potential risks of HVLA in post-osteomyelitis patients (e.g., structural fragility, recurrence risk, etc). Please include a paragraph on safety considerations for HVLA in patients with a history of spinal infections, even after clearance, and discuss potential biomechanical or neurophysiological mechanisms of HVLA in post-osteomyelitis cases.
Response#6: Thank you for this wonderful addition to the discussion. We agree this should be in the discussion section. It has been added and it highlighted in yellow near the end of the discussion section for your convenience.
Comment#7: The role of scapular stabilization exercises and therapeutic alliance is emphasized, but their contributions to outcomes are not parsed. Also, the contributions of HVLA versus scapular stabilization exercises should be differentiated from the observed improvements
Response#7: Thank you for pointing this out. We feel the differentiation is difficult and is address in the now limitations section.
Comment#8: Give us an overview of the weaknesses of your case report
Response#8: Thank you for pointing out that this section is missing from the discussion section. A weaknesses paragraph has been added to the end of the discussion section and highlighted in yellow for your convenience.
Comment#9: Number references in the order in which they are first mentioned in the text. Please arrange the references in the correct format.
Response#9: The references are now in the correct format.
Comment#10: Figure 1 (Graphic 1) is a repetition of Table 6; you should keep either Table 6 or the figure. I suggest keeping the table
Response#10: Thank you for pointing this out. The figure has been removed and the table is now standalone and the phrase "Graphic 1" in the treatment section of the manuscript as been removed.
Reviewer 3 Report
Comments and Suggestions for Authors
The article discusses a relatively rare condition - thoracic spine osteomyelitis and highlights the innovative use of high-velocity low-amplitude (HVLA) rib manipulations, offering novel insights into management strategies. The detailed patient history, clinical findings, and progression of interventions are clearly documented, making it easy to understand and potentially reproducible. The use of standardized outcome measures, such as the Numeric Pain Rating Scale (NPRS), Neck Disability Index (NDI), and Patient-Specific Functional Scale (PSFS), provides clear and quantifiable results that enhance the validity of the findings. The therapeutic progression is explicitly described, particularly the phased integration of manual therapy, therapeutic exercise, and the subsequent introduction of rib manipulation, which increases the clinical applicability of the intervention. The inclusion of a 6-month follow-up, showing sustained improvements and a full return to functional activities, significantly strengthens the clinical impact of the case report. However, as a case report, the inherent limitation is its generalizability; the broader applicability of the results requires cautious interpretation. Without comparative or control data, it is difficult to definitively attribute improvements solely to the intervention. Ethical considerations are properly addressed, with clear patient consent obtained and documented. The case illustrates the potential role of HVLA rib manipulation in managing thoracic spine pain caused by osteomyelitis, suggesting potential broader implications for similar patient populations unresponsive to conventional non-thrust mobilization. The manuscript is well-written and clear. However, initial therapeutic approaches showed minimal improvement, indicating a potential limitation in the earlier treatment selections that should be discussed more thoroughly. A deeper exploration of the pathophysiological mechanisms underlying the efficacy of rib manipulation in this specific scenario would strengthen the manuscript. The methodological rigor is appropriate for a case report, but further details regarding the specific techniques of rib manipulation—such as positioning, force applied, and patient response—would enhance clinical replication. The discussion effectively situates the findings within existing literature but could further emphasize potential physiological rationales, including the mechanisms of rib manipulation that influence thoracic pain and function.
Author Response
Comment#1: The article discusses a relatively rare condition - thoracic spine osteomyelitis and highlights the innovative use of high-velocity low-amplitude (HVLA) rib manipulations, offering novel insights into management strategies. The detailed patient history, clinical findings, and progression of interventions are clearly documented, making it easy to understand and potentially reproducible. The use of standardized outcome measures, such as the Numeric Pain Rating Scale (NPRS), Neck Disability Index (NDI), and Patient-Specific Functional Scale (PSFS), provides clear and quantifiable results that enhance the validity of the findings. The therapeutic progression is explicitly described, particularly the phased integration of manual therapy, therapeutic exercise, and the subsequent introduction of rib manipulation, which increases the clinical applicability of the intervention. The inclusion of a 6-month follow-up, showing sustained improvements and a full return to functional activities, significantly strengthens the clinical impact of the case report. However, as a case report, the inherent limitation is its generalizability; the broader applicability of the results requires cautious interpretation. Without comparative or control data, it is difficult to definitively attribute improvements solely to the intervention. Ethical considerations are properly addressed, with clear patient consent obtained and documented. The case illustrates the potential role of HVLA rib manipulation in managing thoracic spine pain caused by osteomyelitis, suggesting potential broader implications for similar patient populations unresponsive to conventional non-thrust mobilization. The manuscript is well-written and clear. However, initial therapeutic approaches showed minimal improvement, indicating a potential limitation in the earlier treatment selections that should be discussed more thoroughly. A deeper exploration of the pathophysiological mechanisms underlying the efficacy of rib manipulation in this specific scenario would strengthen the manuscript. The methodological rigor is appropriate for a case report, but further details regarding the specific techniques of rib manipulation—such as positioning, force applied, and patient response—would enhance clinical replication. The discussion effectively situates the findings within existing literature but could further emphasize potential physiological rationales, including the mechanisms of rib manipulation that influence thoracic pain and function.
Response#1: Thank you for this detailed comment. The authors have addressed all of the following concerns within other reviewers comments except for the HVLA technique. The technique that was used was added at the beginning of the Treatment section and is highlighted in yellow for your convenience.
Reviewer 4 Report
Comments and Suggestions for Authors
Dear Editor,
Thank you for kindly inviting me to review the manuscript titled “Pain and Disability Reduction Following Rib Manipulation in a Patient Recovering from Osteomyelitis of the Thoracic Spine.”
The authors aim to present a case report on a manipulation-based rehabilitation program applied following vertebral osteomyelitis. Below are my suggestions regarding the manuscript:
Abstract
- Remove the word "Title" from the title and consider adding “case report” to clarify the nature of the study.
- In the abstract, the sentence “At initial evaluation, the patient’s pain intensity score was 8/10 (NPRS, 0-10), disability score was 46/50 (NDI, 0-50), and functional scale score was 3/10 (PSFS, 0-10)” should be rephrased for better readability.
- Select keywords based on MeSH terms.
Introduction
- Review the placement of references throughout the manuscript.
- Ensure the references are formatted according to the journal's guidelines:
- References must be numbered in the order of appearance in the text (including table captions and figure legends) and listed individually at the end of the manuscript.
- In the text, reference numbers should be placed in square brackets [ ], and positioned before punctuation; for example, [1], [1–3], or [1,3].
General Comments
- Double spaces between words in the manuscript should be corrected.
- Review the use of punctuation.
- Provide a brief explanation of how the gait analysis was conducted.
- Indicate the timing of the evaluations conducted before and after treatment.
Discussion
- Strengthen the discussion by referencing the current literature.
Additional Suggestions
- The information presented in Table 4 is general and may not significantly contribute to the manuscript. The authors might consider removing this table to reduce the overall number of tables.
- Although the title emphasizes the reduction of pain and improvement in functionality following rib manipulation, the patient underwent several different treatment modalities. Therefore, attributing the observed improvements solely to manipulation might be misleading. To make such a claim, a comparative approach or session-by-session evaluations would have been necessary. Thus, I recommend revising the title accordingly.
Author Response
Comment#1: Remove the word "Title" from the title and consider adding “case report” to clarify the nature of the study
Response#1: The word "Title" was a place holder on the template and was removed from the manuscript
Comment#2: In the abstract, the sentence “At initial evaluation, the patient’s pain intensity score was 8/10 (NPRS, 0-10), disability score was 46/50 (NDI, 0-50), and functional scale score was 3/10 (PSFS, 0-10)” should be rephrased for better readability.
Response#2: Thank you for pointing this out. The sentence was changed to be more reader friendly and highlighted in yellow for your convenience.
Comment#3: Select keywords based on MeSH terms.
Response#3: All keywords now align with NLM MeSH terms and are highlighted in yellow at the end of the abstract for your convenience.
Comment#4: Review the placement of references throughout the manuscript. Ensure the references are formatted according to the journal's guidelines. References must be numbered in the order of appearance in the text (including table captions and figure legends) and listed individually at the end of the manuscript. In the text, reference numbers should be placed in square brackets [ ], and positioned before punctuation; for example, [1], [1–3], or [1,3].
Response#4: Thank you. This was corrected and now is properly formatted for this journal.
Comment#5: Double spaces between words in the manuscript should be corrected.
Response#5: Thank you, this has been corrected on the word version. There was some difficulty when the journal switches it from my word document to the journal document template but it should be corrected now.
Comment#6: Review the use of punctuation
Response#6: All punctuation was reviewed at anything that was incorrect was fixed
Comment#7: Provide a brief explanation of how the gait analysis was conducted
Response#7: Thank you for pointing out this edit. The sentence was enhanced to discuss further how the gait analysis was performed and is highlighted in yellow for your convenience.
Comment#8: Indicate the timing of the evaluations conducted before and after treatment.
Response#8: Thank you for this comment. The timing is in the tables at the end of the manuscript as well as in the detailed case presentation section
Comment#9: Strengthen the discussion by referencing the current literature
Response#9: This has been done and the discussion section is now enhanced and highlighted in yellow for your convenience.
Comment#10: The information presented in Table 4 is general and may not significantly contribute to the manuscript. The authors might consider removing this table to reduce the overall number of tables.
Response#10: Thank you for this recommendation. We the authors feel that it is important to have the special test explained for those who don't understand what was being performed. For this reason, we will keep Table 4.
Comment#11: Although the title emphasizes the reduction of pain and improvement in functionality following rib manipulation, the patient underwent several different treatment modalities. Therefore, attributing the observed improvements solely to manipulation might be misleading. To make such a claim, a comparative approach or session-by-session evaluations would have been necessary. Thus, I recommend revising the title accordingly
Response#11: Thank you for this comment. The paper has been enhanced significantly while going through the editing process and the title is now appropriate for the way the information is no presented in the manuscript